# Laboratory Analysis of an Asphalt Mixture Overlay Reinforced with a Biaxial Geogrid

Eduardo J. Rueda [1,*], Juan Gabriel Bastidas-Martínez [2], Juan Carlos Ruge [3], Yeison Alayón [2] and Jeisson Olivos [2]

1    Escuela de Construcción Civil, Facultad de Ingenieria, Pontificia Universidad Católica de Chile, Santiago de Chile 7820436, Chile
2    Programa de Especialización en Ingenieria de Pavimentos, Facultad de Ingenieria, Universidad Católica de Colombia, Bogotá 111711, Colombia
3    Programa de Ingenieria Civil, Facultad de Ingenieria, Universidad Católica de Colombia, Bogotá 111711, Colombia
*    Correspondence: eduardo.rueda@uc.cl

**Abstract:** Geosynthetic materials have been demonstrated to be an accurate element in civil engineering, specifically in the field of pavement. Regarding the implementation of these materials in asphalt mixture layers, geosynthetics, such as geotextile and geogrid, have been used to delay the crack propagation and/or increase the fatigue life. However, the use of these material is based on the experience learned in the field or results obtained from testing on a reduced scale in the laboratory. This research work aims at evaluating the influence that geogrids have as a reinforcement to asphalt mixture layer samples. Within this context, in this work, two types of large-scale samples (with and without reinforcement) are subjected to a monotonic load under two support conditions (simple support and granular base). The results were summarized with the load-line displacement diagram, where parameters such as the peak load, displacement, stiffness, and work of fracture were analyzed. The results reveal that the asphalt layer with geogrid experiences a double benefit since it withstands greater magnitudes of load and delays the appearance of rutting problems. To conclude, the geogrid as a reinforcement for asphalt mixture layers strongly impact their mechanical behavior, increasing the service life of the pavement structure.

**Keywords:** geogrid; asphalt mixture; large-scale sample; reinforcement

## 1. Introduction

Asphalt mixture is a widely used material for road construction. To be more precise, between 2017 and 2018, about 4.5 million kilometers (2.8 million miles) were paved in the United States, with 94% using asphalt mixture as the material for the construction of the wearing course [1]. In addition, a critical situation has been emerging with crude oil (the material from which asphalt originates), where the world is facing a shortage. For these reasons, today there is a vast number of investigations that have concentrated their efforts on improving the characteristics of asphalt in order to prolong the useful life of this material and its derivatives. Regarding the techniques used to improve the characteristics of asphalt, the modification with polymers, such as styrene-butadiene rubber or ethylene-vinyl acetate [2,3], biomaterials [4,5], and/or chemical products [6,7] have been extensively used. They have shown favorable results by reducing problems in asphalt mixtures, such as fatigue, rutting, and/or moisture damage. However, the modification of asphalt is not the only strategy used to improve the performance of the asphalt mixture. There is also the use of geosynthetics (i.e., geotextiles, geogrids, and geocells, among others) which have been more focused on granular materials used in paving, such as base, subbase, and subgrade [8].

Concerning the use of geosynthetics as part of the asphalt mixture, several applications exist and work differently within the asphalt mixture depending on the type. Geotextiles

(woven or non-woven) impregnated with an asphalt emulsion have been used to prevent crack propagation occurring from an existing layer to an overlay, and have even been used as a barrier by preventing a passage of water presenting itself in the internal layers of the pavement structure [9]. The correct installation of this material brings favorable results; however, its inappropriate implementation harms the performance of the asphalt mixture [10], hence the importance of large-scale experiments. Although the geotextiles can function as a reinforcement for an asphalt mixture, geogrids are much more effective, improving their behavior against bending and increasing their resistance to tension. This makes the asphalt mixture increase its fatigue life, as well as prevent crack propagation [11]. Most of this evidence was determined through a small-scale laboratory model due to the ease of the specimen's manufacturing, which implies a greater number of tests carried out, and an adequate size for the capacity of most laboratory equipment. Regarding large-scale experimental procedures, there is a limited number of investigations focused on paving, for example C. Zhang et al. [12] carried out an experiment in the field, in which the instrumentation of two sections reinforced with a geogrid was carried out in the embankment foundation system to be monitored and the load transfer capacity promoted by the reinforcement was evaluated. On the other hand, Chen et al. [13] evaluated the structural contribution of the geogrid as a reinforcing element of an embankment. The majority of efforts have been carried out in granular materials, as previously mentioned, while the information on investigations that have used large-scale experimentation to evaluate the performance of geosynthetics in the asphalt mixture is scarce [14].

Based on the aforementioned reasons, this document aims to analyze and quantify the influence that a biaxial geogrid as a reinforcement element has on an asphalt mixture layer. In order to achieve this, test samples were fabricated by trying to approximate the construction procedures in the field and were subjected to a monotonic load under two types of support scenarios. This contributed to the information concerning the large-scale experimental work since this field is expanding and could be considered as a basis in the area of geogrids used in asphalt mixtures, due to its simplicity and low-cost tests that allow the performance of materials in pavement engineering to be evaluated.

## 2. Materials and Experimental Procedure

In this work, the influence that a geogrid has as a reinforcement in an asphalt mixture layer was evaluated. In order to assess the influence of the geogrid, some asphalt mixture specimens were fabricated (i.e., one with the geogrid and another without it as a reinforcement) and evaluated under monotonic load in two support conditions (i.e., with and without a granular material layer that simulated a base layer in a pavement structure).

The following sections describe the characteristics of the materials, the specimen fabrication procedure, and the experimental methodology applied.

### 2.1. Material Properties and Specimen Characteristics

The test sample consists of a slab of asphalt mixture that was designed according to the Marshall mix design method [15]. The asphalt mixture named HMA-12, based on the requirements demanded by the urban development institute of Bogota City (IDU), consisted of a dense-graded HMA with 5.50% asphalt binder content by total weight, an air void content of 5.40%, and a gradation with a nominal maximum aggregate size (NMAS) of 12.5 mm or 1/2 inch (Figure 1) that all correspond to the mean value between the standard specification limits [16]. The asphalt binder was classified as penetration 60–70 [1/10 mm] and the aggregates were of sandstone type.

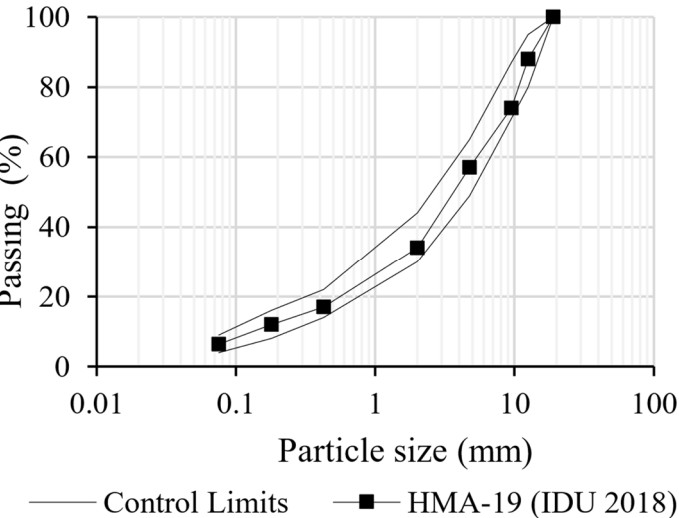

**Figure 1.** Particle size distribution curve for the asphalt mixture.

This asphalt mixture was mechanically evaluated to guarantee an adequate performance behavior. Characterization was achieved through the evaluation of the Marshall stability and flow test [17], the indirect tensile strength test [18], and the tensile strength ratio or TSR [19]. Table 1 summarizes the main results of the asphalt mixture characteristics used to fabricate the test specimens. These results show that this asphalt mixture aims at the requirements demanded, being suitable for use in this research project.

**Table 1.** Asphalt mixture characteristics.

| Characteristic | Result | IDU Specification |
|---|---|---|
| Asphalt Content (%) | 5.5 | - |
| Air voids | 5.4 | 4.0–6.0 |
| Marshall Stability (kN) | 14.82 | ≥9.0 |
| Marshall Flow (mm) | 3.5 | 2.0–3.5 |
| Stability/Flow (kN/mm) | 4.23 | - |
| Indirect Tensile Strength ITS at 25 °C (kPa) | 741 | - |
| Tensile Strength Ratio TSR (%) | 85.2 | ≥80 |

The geogrid was Fortgrid asphalt 700 provided by Geomatrix that consisted of a biaxial grid made from polyethylene terephthalate (PET). This material was located between the two types of asphalt mixtures as a reinforcement to the asphalt layer. Table 2 summarizes the technical characteristics of the geogrid used.

**Table 2.** Physical characteristics of the geogrid.

| Characteristic | Specification |
|---|---|
| Reflective cracking control factor | 7.0 |
| Melting point (°C) | 240 |
| Aperture dimensions (mm) | 20 × 20 |
| Flow Marshall F (mm) | 2.0–3.5 |

As previously mentioned, the test samples consisted of an asphalt mixture sample fabricated in two parts with the same asphalt mixtures. First, the asphalt mixture located at the bottom represented an old degraded asphalt mixture, and for this reason, at the lowest part of the layer, a notch of 3 mm in length and 20 mm in depth was made using a saw. The purpose of this notch was to weaken the mixture with a crack, as a result of the repetitive passage of the vehicles. The second part was located at the top part of the layer (below

the asphalt mixture with the notch) and represented a new asphalt mixture placed as an overlay. The geogrid was located between the two asphalt mixtures. For each of the asphalt mixtures, the geometrical configuration of the test sample was a square slab with a side length of 30 cm and a thickness of 5 cm (Figure 2).

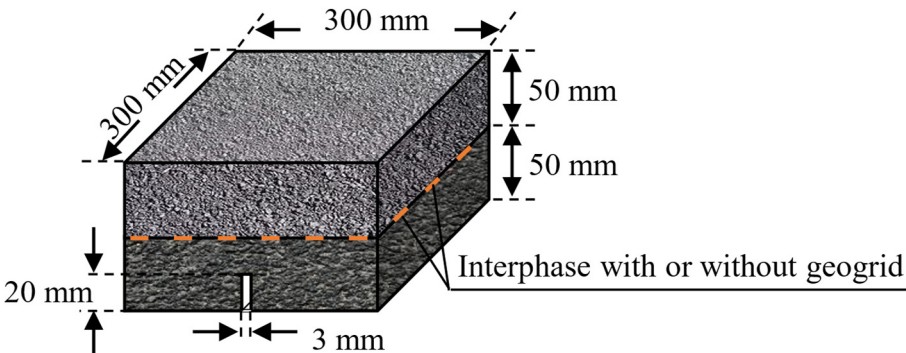

**Figure 2.** Geometrical configuration of the test sample.

The fabrication of the test samples was carried out in the field. This process was initiated with the demarcation of a 2 m² work area through the use of metal forms. The first part of the asphalt mixture was placed and compacted inside the forms, with a thickness of 5 cm (Figure 3a). After that, a prime coat with a cationic rapid setting asphalt emulsion (CRR-1 according to IDU) was applied to the surface of the compacted asphalt mixture and the geogrid was then placed over half of the surface (Figure 3b). Next, the second asphalt mixture layer of 5 cm was compacted on the entire surface of the work area. The compaction process was carried out on an asphalt mixture with a temperature range between 140 and 145 °C, with a roller compactor, that applied a static load of 5 t and passed over the mixture until it fit the mold (Figure 3c,d). Finally, the asphalt mixture slabs were cut into 30 cm side squares, and at the bottom of the test sample, a notch was made. In total, 18 test samples were obtained (9 for each reinforcement condition).

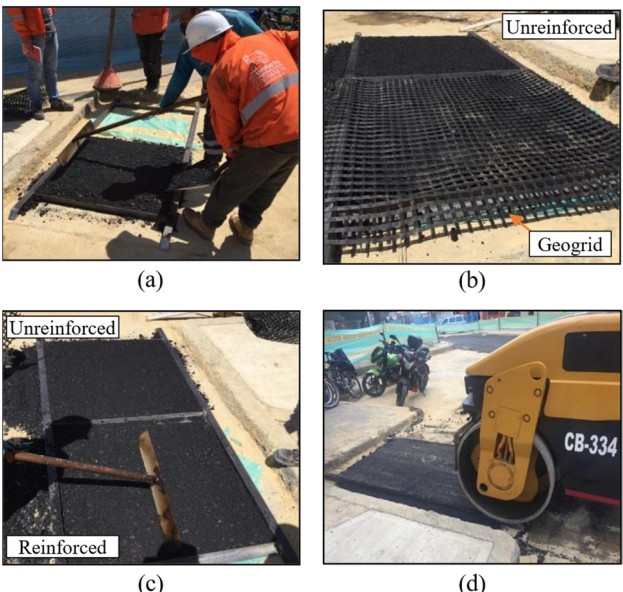

**Figure 3.** Test sample fabrication process: (**a**) area demarcation, (**b**) location of the geogrid, (**c**) the second asphalt layer, and finally (**d**) the compaction process.

## 2.2. Experimental Procedure

Before the experimental methodology to determine the influence of the geogrid within the asphalt mixture was performed, the volumetric properties, as the theoretical maximum

specific gravity [20] of the HMA, and the bulk specific gravity [21] of the testing specimens were determined in order to control the air void content. Once the testing was completed, the testing specimens were used in an experimental procedure that consisted of subjecting the specimens to a compaction monotonic load until they reached the peak load. This methodology was applied under two scenarios related to the support condition of the sample (Figure 4a). In the first setup, the specimen was supported as a simple beam with the monotonic load applied at the center, while the second scenario considered the use of a granular material as a support for the sample, simulating a base layer (Figure 4b). It is noteworthy that each scenario pretended to represent a performance condition of the asphalt mixture in a pavement structure.

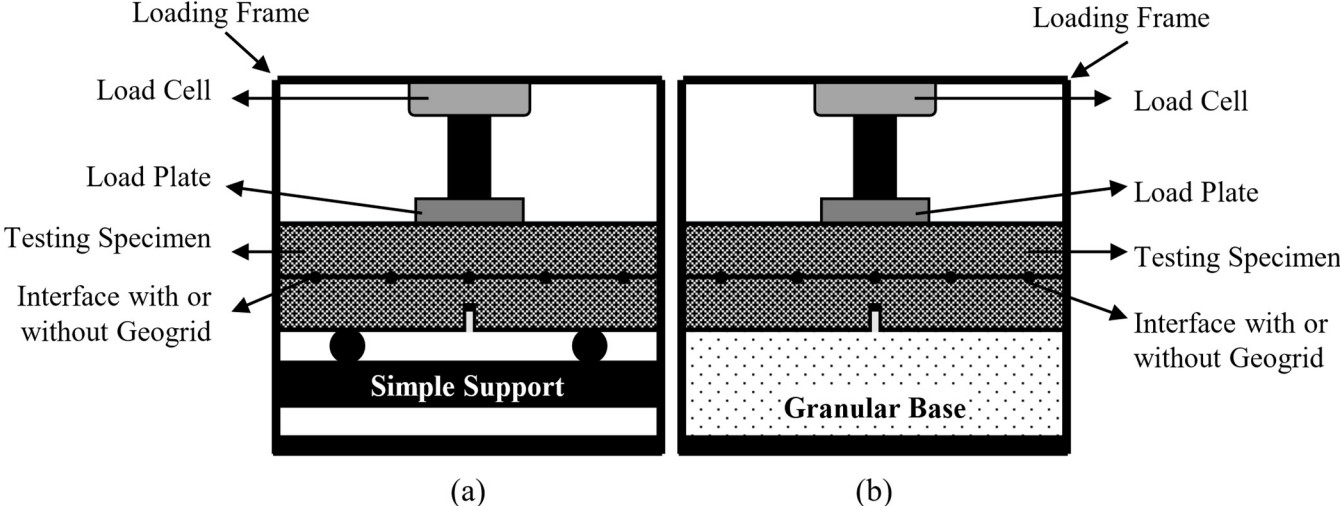

**Figure 4.** Loading application sketch of the two scenarios (**a**) with simple support, and (**b**) a granular base as a support.

In terms of the granular material, a GB-38 type was utilized. According to the IDU requirements, this material had an NMAS of 19 mm or $\frac{3}{4}$ inches (Figure 5) gradation distribution and was characterized considering four main aspects. The first aspect was the hardness, where the Los Angeles Abrasion Test [22] and the testing aggregates methods for the determination of the aggregate crushing value [23] were applied to evaluate the durability. Cleanliness was the second aspect considered for the aggregate evaluation. In this aspect, test methods to determine the liquid limit [24], the plasticity index [24], and the sand equivalent value [25] were carried out. The aggregate particle shape index [26] was used to evaluate the geometric particle configuration. The last aspect was the capacity of the granular material which was evaluated through the proctor compaction test [27] and the standard test method for the California Bearing Ratio (CBR). Table 3 shows the current values and the requirements.

The compaction process of the material was conducted on a metallic box, where the two granular material layers were placed until reaching a base of 20 cm height. The material was densified with the maximum unit weight and the optimum moisture content being based on the results obtained from the proctor compaction test [27]. Once this process was finished, it was ready to be the support for the asphalt mixture testing sample. It is worth noting that for each test sample evaluated, a granular base was compacted.

The experimental procedure consisted of applying a monotonic load using Material Testing System equipment (MTS Exceed model e45, MTS Systems Corporation, Eden Prairie, MN, United States) at a loading rate of 1 mm/min, under displacement control. The load was applied to the testing samples through a metallic circular plate of 10 cm diameter, representing an approximation of the contact area between the tire and the pavement surface. To measure the load applied by the equipment and the displacement development in the testing sample, a load cell was placed at the top part of the loading actuator, and

a linear variable differential transformer (LVDT) was placed in the middle of the testing sample. The temperature of the asphalt mixture specimens was 21 ± 1 °C.

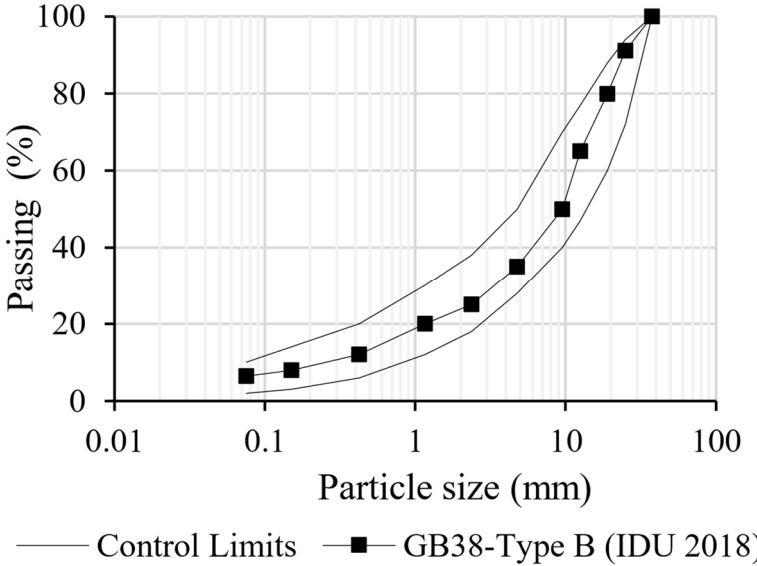

**Figure 5.** Base Granular material size distribution.

**Table 3.** Granular material properties.

| Test | Result | Requirement |
|---|---|---|
| Hardness | | |
| Los Angeles abrasion resistance (%) | 38.9 | max 40.0 |
| Aggregate crushing value (kN) | 171 | min 60 |
| Aggregate crushing ratio wet/dry (%) | 85 | min 75 |
| Cleanliness | | |
| Liquid limit (%) | 0 | max 25 |
| Plasticity index (%) | 0 | max 3 |
| Sand equivalent value (%) | 49 | min 55.0 |
| Geometry particle configuration | | |
| Aggregate particle shape index (-) | 0.7 | min 0.5 |
| Capacity | | |
| Maximum dry unit weight (kN/m$^3$) | 20.3 | - |
| Optimum moisture content (%) | 8.1 | - |
| CBR Value (%) | 87 | >80 |

In general, a total of twelve specimens were tested. Considering that there were two reinforcement conditions, two support scenarios and three replicates under each condition and scenario were subjected to the experimental procedure.

- HMA Unreinforced—Simple support (USS);
- HMA Reinforced—Simple support (RSS);
- HMA Unreinforced—Granular base support (UGS);
- HMA Reinforced—Granular base support (RGS).

It is important to mention that even though the load applied on a pavement structure is dynamic, the monotonic loading scheme allows the evaluation of the geogrid performance as a reinforcement element in an asphalt mixture. Regarding the support for the testing specimens, this research aimed to evaluate the performance of the geogrid. On the one hand, the geogrid underwent an extreme condition when the evaluation of the properties considered the simple support, just because the geogrid was forced to work at its maximum

capacity, while on the other, a granular base was used as a support structure to create a more realistic condition. In this case, the granular base allows the mobilization of the stresses coming from the asphalt mixture layer. In this way, the proposed experimental methodology allows the comparison between the reinforced and non-reinforced asphalt layers. As an example, Figure 6 presents the experimental setup for the unreinforced HMA with a granular base support scenario and the final result after subjecting the testing sample to the monotonic load.

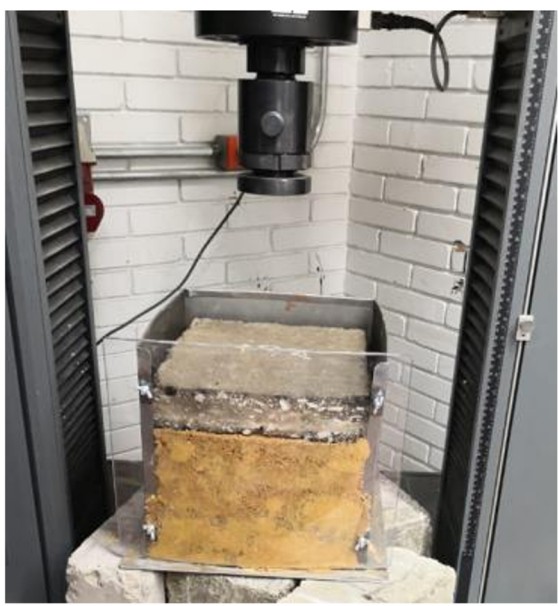

**Figure 6.** Geometrical configuration of the testing specimen with the granular base as a support.

## 3. Results and Analyses

The data obtained after subjecting the testing specimens to the experimental procedure consisted of load and displacement parameters. This information was used to plot the load–load-line displacement diagram (LLD) (Figure 7) [28], where important information on the performance behavior could be obtained, such as the peak load (the maximum load develops for the material until it reaches rupture), the load-line displacement (deformation at the failure point), the stiffness (the ratio between the reaction forces and the produced deflection, before the peak load), and the work of fracture (the area under an LLD graph).

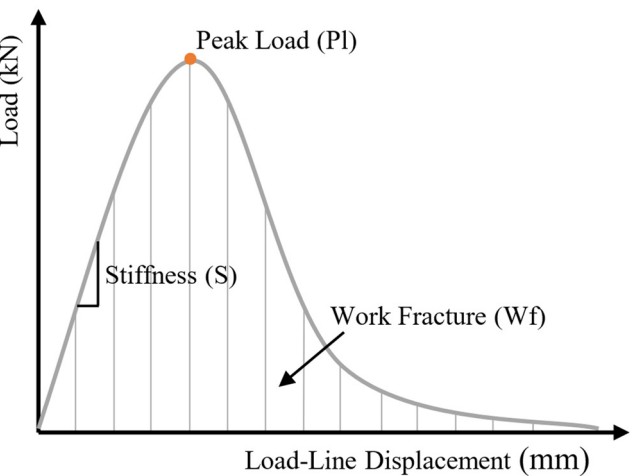

**Figure 7.** Theoretical load–load-line displacement diagram.

According to Figure 8, it is evident that the weakest scenario is the asphalt mixture without reinforcement that was tested under a simple support condition, while the most

favorable is the reinforced testing sample with the granular bases as a support, with the peak load developing 28.2 times larger (i.e., 1.12 and 31.63 kN, respectively). In general, the reinforcement in the samples and the test support conditions influenced the mechanical performance of the asphalt mixture.

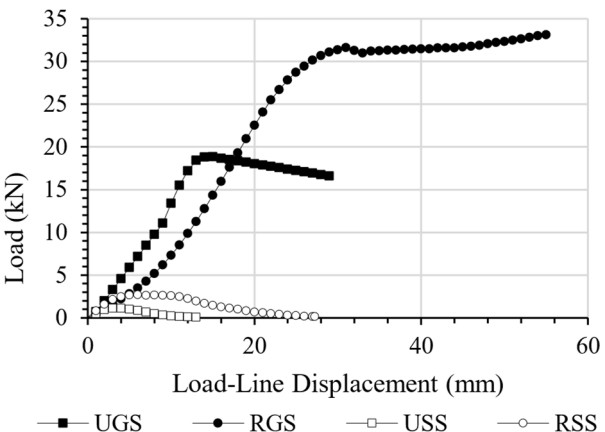

**Figure 8.** Load-line displacement diagram for all scenarios.

Furthermore, the reinforcement helped the asphalt mixture to develop a more significant peak load. In the case of the testing samples that were tested on the simple support, the maximum load developed by the reinforced mixtures was 2.70 kN, while this magnitude without reinforcement was 1.12 kN, showing an increase of 140% or 2.4 times. This increase for the samples that were tested using the granular base as a support was slightly better, but still significant since the peak load increased by 67.9% in the reinforced samples compared to those that were not reinforced (Figure 9a). The reason that tries to explain why the increase in load was greater when the test was performed on the samples under the simple support condition is that, under this condition, the geogrid had the opportunity to work at 100%. In the case of the samples that were supported on the granular base, this type of support helped in the development of the load, not allowing the geogrid to be stressed to the limit.

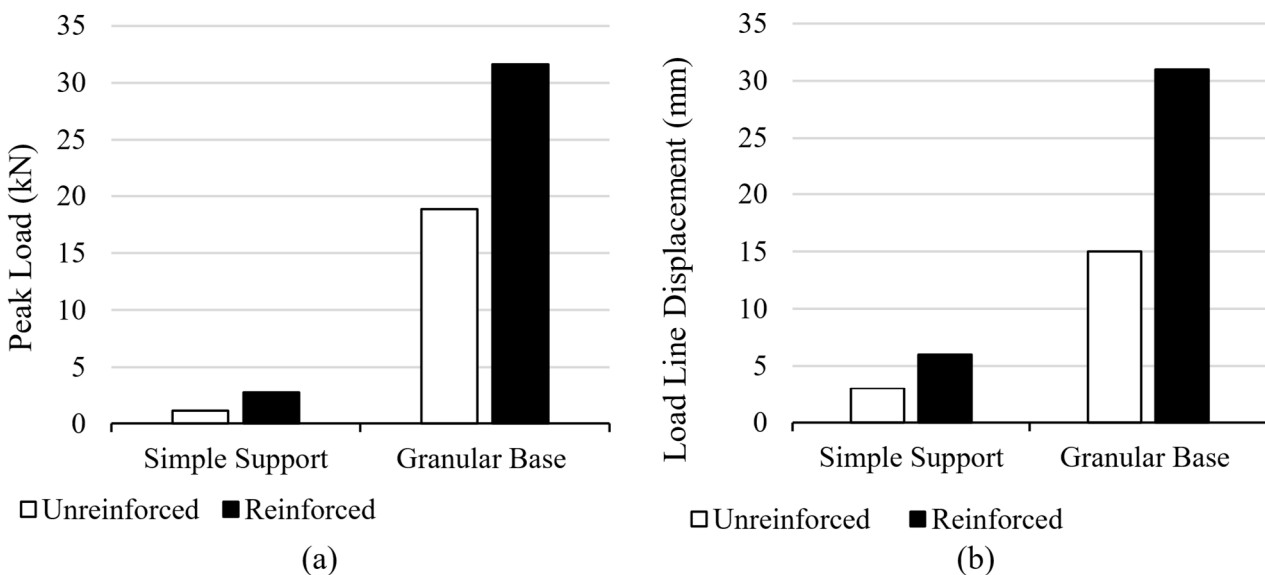

**Figure 9.** Comparison between the results obtained from (**a**) peak load and (**b**) load-line displacement for the testing samples.

Concerning the load-line displacement, reinforcement helped the tested samples to develop a greater displacement before reaching failure, thus helping the reinforced samples

to double the magnitude of those without reinforcement. In the case of the samples tested in a simple support condition, the samples without reinforcement presented a deformation of 3 mm, and of 6 mm for those with reinforcement, thus showing an increase of 100%. The same scenario happened under the granular base scenario, where the contribution was 106.7%. At this stage, in terms of magnitude, the displacement developed by the samples supported on the granular base was more remarkable, and the increased contribution of the reinforcement did not have a greater difference between the support conditions, being around 100% (Figure 9b). Another aspect in which the geogrid had an influence was in the discharge speed, where it occurred much faster in those samples without reinforcement (Figure 8). This is because once the asphalt mixture reached the fault, the geogrid assumed the load and released it more smoothly.

Considering that, by definition, the stiffness of the material is the ability to resist deformation when an external force is applied [29], for the samples tested in the simple support condition, it was observed that the use of the geogrid increased the stiffness of the asphalt mixture specimen by 20.1%, allowing the mixture to support a higher load and develop less deformation. However, the opposite occurred with the reinforced asphalt mixture tested on the granular base, since the stiffness of the sample decreased by 18.8% compared to the sample tested under the same conditions but without reinforcement (Figure 10a). This was an unexpected result and could be explained by the RGS-conditioned sample developing a load greater than 30 kN and a displacement of 30 mm, which can be considered large magnitudes. This was due to the contribution of the geogrid and the granular base that allowed the system to present a ductile behavior. That is, after failure, the geogrid allowed the load to be maintained, while the granular base distributed the load to achieve a greater deformation. This is the reason why stiffness decreases. This behavior was visually evidenced in the laboratory, as shown in Figure 11. Even with this peculiar behavior, this sample was the one that performed better. In addition, it was the one that developed the highest work of fracture, meaning that more energy was required to be able to break the samples reinforced with the geogrid (Figure 10b). Under both scenarios, the reinforced asphalt mixtures presented a higher work of fracture, 39.93 and 1252.5 kg·m$^2$/s$^2$, respectively, with the developed magnitudes being higher when the samples were tested on the granular base. As mentioned before, the granular base provided a higher stability to the samples and indirectly strengthened the system.

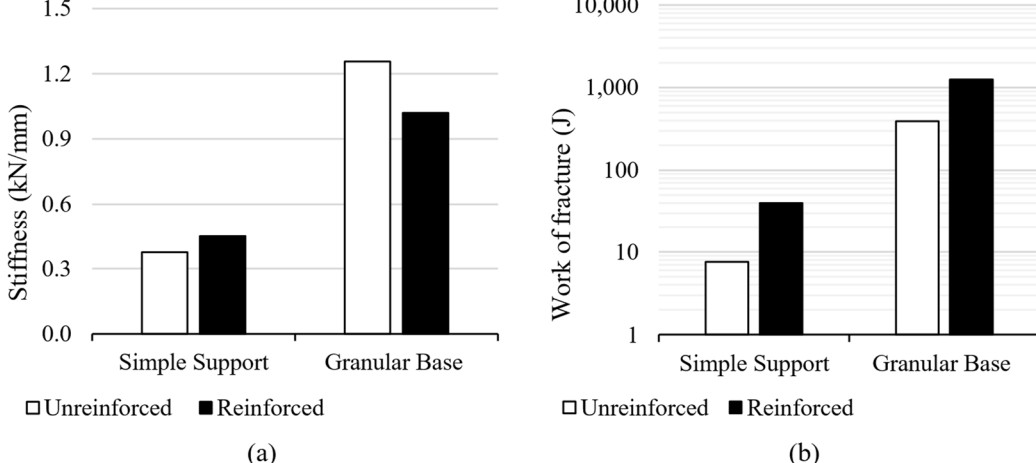

**Figure 10.** Comparison between the results obtained from (**a**) stiffness and (**b**) work of fracture in a log scale from the testing samples.

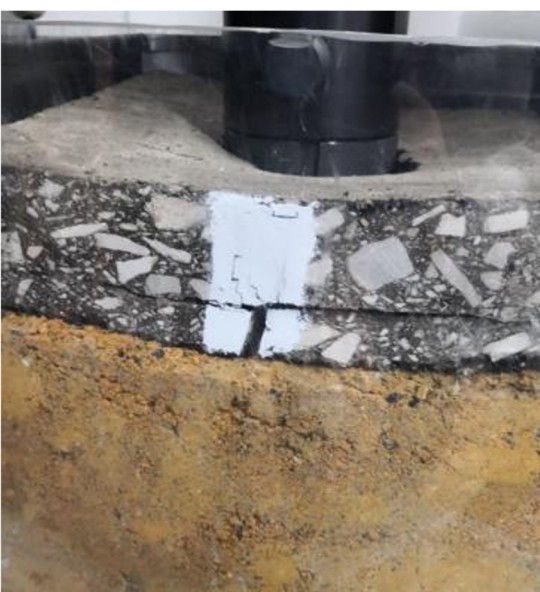

**Figure 11.** Failure mechanism after testing the specimen with the granular base as a support.

## 4. Conclusions

This research aimed to quantify the influence of a geogrid as a reinforcement material in asphalt layers. The asphalt mixture samples consisted of two layers of the same asphalt mixture, each asphalt layer being 5 cm in thickness, with the possibility of placing the geogrid as a reinforcement between them. To achieve the objective, these testing samples were evaluated by subjecting them to a monotonic load under two support stages (i.e., simple support and/or granular base).

The result was summarized with the load-line displacement diagram, where information such as the peak load, load-line displacement, stiffness, and the work of fracture were obtained and analyzed to determine the influence of the geogrid. In summary, even though it is known that the test conducted under simple support is not a real condition for an asphalt mixture layer, it permits the quantification of the effect of the geogrid as part of the layer. In terms of the peak load, the use of the geogrid increased this magnitude by 140%. This increase was not so important when the test was conducted on the granular base support. Although the magnitude increased between 1073.4% and 1579.3% due to the support provided by the granular material, an increment resulting from the geogrid was 69%, just because the reinforcement did not work at its maximum capacity. Regarding the load-line displacement, the use of the reinforcement caused the sample to develop a more significant deformation without failure. It is noteworthy that rutting is an aspect that depends on the asphalt properties, the mixture design method, and the support condition. However, the geogrid reduced the displacement growth rate, meaning that at the same load, the reinforced test sample developed less deformation than the test sample without reinforcement. Thus, the use of a geogrid helps to extend and lengthen the time of appearance of permanent deformation.

The reinforced samples proved to have a greater stiffness, that is, under the same level of deformation, the samples managed to develop a greater load. In addition, they also gained ductility, since these samples managed to develop greater deformations for failure to occur. In this way, it can be evidenced that the use of the geogrid as a reinforcement element in asphalt layers brings a double benefit since it withstands greater magnitudes of load and, at the same time, increases the deformation developed, or, based on what has been previously mentioned, delays the appearance of rutting problems. In conclusion, it increases the ability to absorb energy before breaking (work of fracture), delaying damage to the structure.

To conclude, the results demonstrated the use of a geogrid as a reinforcement of asphalt mixture layers and hence its mechanical performance. To continue extending this knowledge, it is important evaluate the mechanical behavior under dynamic loads of these large-scale samples.

**Author Contributions:** J.G.B.-M., E.J.R. and J.C.R. conceived and designed the research; Y.A. and J.O. executed the methodology; E.J.R. and J.G.B.-M. drafted the manuscript, prepared figures and revised the manuscript; J.G.B.-M., E.J.R. and J.C.R. discussed the results. All authors have read and agreed to the published version of the manuscript.

**Funding:** This research received no external funding.

**Institutional Review Board Statement:** Not applicable.

**Informed Consent Statement:** Not applicable.

**Data Availability Statement:** Not applicable.

**Acknowledgments:** This publication was partially made possible by the call for proposals "Fondo de apoyo para la revisión, edición y/o traducción de artículos para envío a revistas científicas de alto impacto indexadas Web of Science/Scopus" and "fondo de apoyo para los cargos de procesamiento de artículos en revistas de científicas alto impacto indexadas Web of Science/Scopus" from The Research and Development Department (DID) of the School of Civil Construction UC (ECCUC) at Pontificia Católica de Chile.To Universidad Católica de Colombia for the support granted to researchers. Its contents are solely the responsibility of the authors and do not necessarily represent the official views of the University.

**Conflicts of Interest:** The authors declare no conflict of interest.

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
