# Peer review of "Laboratory Analysis of an Asphalt Mixture Overlay Reinforced with a Biaxial Geogrid"

_coatings, doi:10.3390/coatings13010099_

Round 1

Reviewer 1 Report

1. Nearly half of the abstract is about the antecedent review (98 words) and the rest is about this work (101). This abstract should be focused on the contribution of this work rather than a review that corresponds to the introduction section. The conclusions section can provide some hints for the abstract.

2. Avoid adjectives such as “good”: “good performance behavior

3. In section 2.1, how much samples were tested?

Line191 state a total of 12 samples.

4. In Figure 1, what means “passing”?

do you mean that the aggregate is within the range? Below the NMAS

5. In Tables 1 and 2, the values columns can be center aligned.

6. Page 4, line 127: Use superscripts in “2 m2” as “2 m2

5 Ton

7. Figure 11. The y-axis level requires a change “stifness”

8. Page 10, line 287: It is preferable “highest” than “greatest”

Page 10, line 291: It is preferable “higher” than “greater”

9. It is unclear if Figures 6 and 10 are without geogrid and require a comparison with other images with the geogrid in position.

Author Response

  1. Nearly half of the abstract is about the antecedent review (98 words) and the rest is about this work (101). This abstract should be focused on the contribution of this work rather than a review that corresponds to the introduction section. The conclusions section can provide some hints for the abstract.

I agree that the abstract was unbalanced, now the abstract is made up of Introduction (74 words) Objective (19 words) Methods (30 words) Results (24 words) Discussion (51 words) for a total of 198 words.

  1. Avoid adjectives such as “good”: “good performance behavior”

The word “good” was replaced by “adequate”

  1. In section 2.1, how much samples were tested?

A total of twelve specimens were tested

Three (3) samples for HMA Unreinforced – Simple support (USS);

Three (3) samples for HMA Reinforced – Simple support (RSS);

Three (3) samples for HMA Unreinforced – Granular base support (UGS);

Three (3) samples for HMA Reinforced – Granular base support (RGS).

  1. In Figure 1, what means “passing”?

do you mean that the aggregate is within the range? Below the NMAS

The vertical axle of the grading curve represents in % the weight of the particles passing through each sieve. For example, in Figure 1, approximately 80% of the material passes through the 10 mm opening sieve.

The nominal maximum aggregate size (NMAS) is defined as "one sieve size larger than the first size to retain more than 10%. So, this value is including the grading curve.

  1. In Tables 1 and 2, the values columns can be center aligned.

The numerical values of tables 1 and 2 were aligned to the center.

  1. Page 4, line 127: Use superscripts in “2 m2” as “2 m2

These typing errors were fixed

  1. Figure 11. The y-axis level requires a change “stiffness”

This typing error was fixed

  1. Page 10, line 287: It is preferable “highest” than “greatest”

Page 10, line 291: It is preferable “higher” than “greater”

The words were replaced, considered the most appropriate

  1. It is unclear if Figures 6 and 10 are without geogrid and require a comparison with other images with the geogrid in position.

I agree, it is difficult to identify in the photographs if it is a sample with or without reinforcement, reviewing the photographs taken during the development of the work, this difference is not easily evident, we know what the photographs in the documents correspond to thanks to the notes, but it is difficult to identify this since the geogrid is inside the tested sample.

Reviewer 2 Report

The study deals with an interesting topic and the following amendments it might be helpful to improve it:

1- The abstract contains some unnecessary information making it bulky the important results should be listed in numbers of percentages in it.

2-Why the air void content was 5.40% any specific reason for that according to the Marshall mix design method it should be between 3-5%.

3- An experimental procedure that consisted of subjecting the specimens to a compaction monotonic load until they reached the peak load. Could the author mention the machine used to apply the load and what was the load applied?is it subjected to any condition or freeload?

4- In Material properties and specimen characteristics, you mentioned that the nominal maximum aggregate size (NMAS) was 12.5 mm while in the Experimental procedure you were explaining (According to the IDU requirements, this material had an NMAS of 19 mm) what is the relation between it or any justification why you are explaining about 19 mm?

Author Response

  • The abstract contains some unnecessary information making it bulky the important results should be listed in numbers of percentages in it.

I agree that the abstract was unbalanced, now the abstract is made up of Introduction (74 words) Objective (19 words) Methods (30 words) Results (24 words) Discussion (51 words) for a total of 198 words.

  • Why the air void content was 5.40% any specific reason for that according to the Marshall mix design method it should be between 3-5%.

Totally agree that the percentage of voids is not adequate, the problem was in the construction process of the samples since we consider that there was an edge effect of the mold. Although we recognize that the percentage of voids is slightly above the value established by the Marshall methodology, it is noteworthy that this study focuses on quantifying the influences that the geogrid has as a reinforcement element of the asphalt layer, so it is possible that this small difference does not have a significant impact on the way the geogrid contributes.

  • An experimental procedure that consisted of subjecting the specimens to a compaction monotonic load until they reached the peak load. Could the author mention the machine used to apply the load and what was the load applied? is it subjected to any condition or freeload?

The equipment used for the application of the load corresponds to a reference press MTS Exceed model e45 and the control of the equipment was by deformation and the load application speed was 5 mm/min. The team name was put in the document and the loading speed corrected.

  • In Material properties and specimen characteristics, you mentioned that the nominal maximum aggregate size (NMAS) was 12.5 mm while in the Experimental procedure you were explaining (According to the IDU requirements, this material had an NMAS of 19 mm) what is the relation between it or any justification why you are explaining about 19 mm?

The document presents two granulometries, one for the asphalt mixture which has a nominal maximum size (NMAS) of 12.5mm and the second granulometry is that of the base granular material which has a NMAS of 19mm. The error in this part was written since the asphalt mixture according to the DOT is called HMA12, while what is written is HMA 19. This error has already been corrected in the document.

Reviewer 3 Report

Although there have been a lot of studies on the application of Geogrid as reinforcement to control the reflective cracks in asphalt pavement overlay, the fabrication method of largescale test specimens in this study is relatively close to the actual pavement construction method, with certain characteristics, but the relevant conclusions lack novelty. Two suggestions:

(1) The "Reflective cracking control factor" in Table 2 lacks an effective definition;

(2) In fact, there have been quite a lot of studies on the reinforcement effect of the geogrid in the asphalt layer and the effect of controlling reflective cracks, but there are few studies on the size effect of the geogrid, such as the length of the geogrid uesed for existing pavement cracks, which will directly affect the technical and economic effects of the geogrid application. If this research can use largescal test specimens to supplement the research data in this regard, it will greatly enhance the value of this research.

Author Response

Although there have been a lot of studies on the application of Geogrid as reinforcement to control the reflective cracks in asphalt pavement overlay, the fabrication method of largescale test specimens in this study is relatively close to the actual pavement construction method, with certain characteristics, but the relevant conclusions lack novelty. Two suggestions:

  • The "Reflective cracking control factor" in Table 2 lacks an effective definition;

I greatly appreciate your comments, as they help improve the quality of the document. In this opportunity The "Reflective cracking control factor" is one of the specifications of the Goematrix brand geogrid and when investigating its definition, this parameter is determined from performing the fatigue test on specimens with and without geogrid reinforcement and quantify the number of cycles necessary for the appearance of the crack in the overlay.

  • In fact, there have been quite a lot of studies on the reinforcement effect of the geogrid in the asphalt layer and the effect of controlling reflective cracks, but there are few studies on the size effect of the geogrid, such as the length of the geogrid used for existing pavement cracks, which will directly affect the technical and economic effects of the geogrid application. If this research can use larg escal test specimens to supplement the research data in this regard, it will greatly enhance the value of this research.

Thank you very much, the intention of this research was to contribute with data obtained in large samples, in which the closest thing to the real way of building roads will be carried out.

Round 2

Reviewer 1 Report

I validate that the changes suggested were answered appropriately.
